# Fibrillar Collagen Type I Participates in the Survival and Aggregation of Primary Hepatocytes Cultured on Soft Hydrogels

**DOI:** 10.3390/biomimetics5020030

**Published:** 2020-06-25

**Authors:** Nathalia Serna-Márquez, Adriana Rodríguez-Hernández, Marisol Ayala-Reyes, Lorena Omega Martínez-Hernández, Miguel Ángel Peña-Rico, Jorge Carretero-Ortega, Mathieu Hautefeuille, Genaro Vázquez-Victorio

**Affiliations:** 1Laboratorio Nacional de Soluciones Biomiméticas para Diagnóstico y Terapia (LaNSBioDyT), Facultad de Ciencias, Universidad Nacional Autónoma de Mexico, Ciudad de México CP 04510, Mexico; nathalia.serna@ciencias.unam.mx (N.S.-M.); adriana.rodriguez@ciencias.unam.mx (A.R.-H.); marisol.ayala@ciencias.unam.mx (M.A.-R.); lorenaomega94@gmail.com (L.O.M.-H.); jorge.carretero@ciencias.unam.mx (J.C.-O.); mathieu_h@ciencias.unam.mx (M.H.); 2Instituto de Biotecnología, Universidad del Papaloapan, Tuxtepec CP 68301, Oaxaca, Mexico; mapena@unpa.edu.mx; 3Departamento de Física, Facultad de Ciencias, Universidad Nacional Autónoma de Mexico, Ciudad de México CP 04510, Mexico

**Keywords:** primary hepatocytes, polyacrylamide hydrogels, stiffness, collagen, basement membrane, actin cytoskeleton, epidermal growth factor (EGF)

## Abstract

Liver is an essential organ that carries out multiple functions such as glycogen storage, the synthesis of plasma proteins, and the detoxification of xenobiotics. Hepatocytes are the parenchyma that sustain almost all the functions supported by this organ. Hepatocytes and non-parenchymal cells respond to the mechanical alterations that occur in the extracellular matrix (ECM) caused by organogenesis and regenerating processes. Rearrangements of the ECM modify the composition and mechanical properties that result in specific dedifferentiation programs inside the hepatic cells. Quiescent hepatocytes are embedded in the soft ECM, which contains an important concentration of fibrillar collagens in combination with a basement membrane-associated matrix (BM). This work aims to evaluate the role of fibrillar collagens and BM on actin cytoskeleton organization and the function of rat primary hepatocytes cultured on soft elastic polyacrylamide hydrogels (PAA HGs). We used rat tail collagen type I and Matrigel^®^ as references of fibrillar collagens and BM respectively and mixed different percentages of collagen type I in combination with BM. We also used peptides obtained from decellularized liver matrices (dECM). Remarkably, hepatocytes showed a poor adhesion in the absence of collagen on soft PAA HGs. We demonstrated that collagen type I inhibited apoptosis and activated extracellular signal-regulated kinases 1/2 (ERK1/2) in primary hepatocytes cultured on soft hydrogels. Epidermal growth factor (EGF) was not able to rescue cell viability in conjugated BM but affected cell aggregation in soft PAA HGs conjugated with combinations of different proportions of collagen and BM. Interestingly, actin cytoskeleton was localized and preserved close to plasma membrane (cortical actin) and proximal to intercellular ducts (canaliculi-like structures) in soft conditions; however, albumin protein expression was not preserved, even though primary hepatocytes did not remodel their actin cytoskeleton significantly in soft conditions. This investigation highlights the important role of fibrillar collagens on soft hydrogels for the maintenance of survival and aggregation of the hepatocytes. Data suggest evaluating the conditions that allow the establishment of optimal biomimetic environments for physiology and cell biology studies, where the phenotype of primary cells may be preserved for longer periods of time.

## 1. Introduction

The liver is an essential organ that carries out multiple functions such as glycogen storage, serum proteins production, and detoxification [1,2]. Hepatocytes are the parenchyma that sustain the functions of the whole organ [1,3]. Non-parenchymal cells maintain the homeostasis of hepatocytes and participate in other physiological processes such as regeneration where the signaling and metabolism of hepatocytes are switched [4,5]. Quiescent hepatocytes are organized in elongated aggregates (hepatic cordons) that form the functional unit known as the hepatic lobule. Inside the hepatic lobules, hepatocytes participate in different functions throughout the hepatic cordons (zonation) [1,2,6]. In addition, hepatocytes construct intercellular ducts (bile canaliculi) which they transport bile salts to outside the liver [2,6]. Bile canaliculi establish the polarity and the direction of vesicle transport inside the hepatocytes [7]. Alterations in bile canaliculi are associated with different physiopathological processes [8,9]. The plasma membrane of the hepatocytes that is not surrounding canaliculi is in direct contact with a soft extracellular matrix (ECM) [10,11]. Hepatic stellate cells (HSCs) and liver sinusoidal endothelial cells (LSECs) share this soft ECM in a perisinusoidal space known as space of Disse [6,12,13]. Inside this space, fibrillar collagens type I and III are connected by collagen type V, forming a reticular conformation [11,14,15]. Collagen type IV is also deposited in the perisinusoidal space, although this ECM protein is not forming a basal lamina as in classic basement membranes [10,16,17,18]. Laminins and fibronectin are not abundant in the space of Disse, but both ECM proteins seem to play an important role in liver organogenesis and the maintenance of progenitors [19,20]. Therefore, hepatocytes are embedded in a reticular ECM with a differential distribution of its main components (zonation), fibrillar collagens, and a basement membrane-associated matrix, and this zonation might be related with the specification of hepatocytes within the hepatic lobule [1,21,22].

It is well known that primary hepatocytes cultured on ECM-coated polystyrene plates transdifferentiate into a mesenchymal-like phenotype after approximately 72 h of culture, losing their typical functions such as albumin secretion, glycogen storage, urea synthesis, drug metabolism, etc. [23,24]. Hence, researchers have tested different culture conditions in order to maintain functional primary hepatocytes during culture, aiming at finding the right combinations that help preserve most of the aforementioned cues in vitro [10,23,25,26,27,28]. Hepatocytes are epithelial cells whose polarization is based on the formation of canaliculi, which are regrettably lost during culture [7,8,29]. Bile canaliculi indeed seem to be important for the proliferation and homeostasis of hepatocytes [8,9], and the culture of primary hepatocytes must then consider that the maintenance of canaliculi or canaliculi-like structures (partial ducts) is a good indicator of the desired phenotype [29,30,31,32]. In addition to canaliculi, the loss of polarity and epithelial phenotype is tightly associated with modifications in actin cytoskeleton in primary hepatocytes. In standard polystyrene culture plates, cuboidal epithelial-like hepatocytes remodel their actin cytoskeleton (actin filaments) from structures that are cortical and proximal to canaliculi toward more anchored stress fibers localized at the periphery of planar mesenchymal-like cells [29,30,33]. Both cortical actin (F-actin) and stress fibers are structures that generate tensile forces [9,33,34], and it has been demonstrated that the increase of cell area in primary hepatocytes is a consequence of an increase in anchoring mediated by integrins and the presence of tensile forces mediated by stress fibers [26,33,35,36,37,38]. Recently, Sun et al. demonstrated that the inhibition of actomyosin-associated forces and mechanotransduction signals preserves many functions of primary hepatocytes on standard polystyrene culture during 3 weeks [33]. In parallel, another group showed similar results by inhibiting some signal transduction pathways [39]. In that report, researchers used forskolin and SB431542 inhibitor, both of which are known to regulate actin cytoskeleton dynamics by inhibiting the effects on actin polymerization that cyclic-AMP phosphodiesterases and transforming growth factor beta (TGF-β) signals increase [40,41].

Cell biologists are currently aware of the great impact of mechanical stress and resultant strains on all cell processes [42,43,44,45]. Cells respond to external mechanical properties by transducing them into biochemical mechanisms; all these molecular mechanisms are known as mechanotransduction pathways [46,47,48,49]. Despite the complexity of mechanotransduction mechanisms, actin cytoskeleton dynamics is involved in most of them, due to the fact that external stress is transmitted as internal tensile forces that totally depend on the actomyosin apparatus: actin fibers and force-generating proteins, myosins [50,51,52]. There is an increasing number of scientific reports demonstrating that in the liver too, hepatocytes and non-parenchymal cells are regulated by external mechanics [26,37,53,54]. In particular, hepatocytes have been cultured on various conditions or platforms with different mechanical properties to preserve a functional phenotype similar to the phenotype reported in the liver because in vitro cultured hepatocytes dedifferentiate to a mesenchymal-like state [25,26,27,28,55]. It was reported that the elastic modulus along the hepatic lobules is around 150 Pa in normal conditions, and it increases up to 3 kPa depending on the fibrotic model [26]; nonetheless, the attenuation of cell spreading has been reported ≥1 kPa in hepatic cells [56]. Fabricating hydrogels based on ECM proteins is the most common strategy to maintain primary hepatocytes in soft conditions with an elastic modulus similar to what is found in liver tissue [10,28,55]. The use of basement membrane matrix extracted from Engelbreth–Holm–Swarm (EHS) mouse sarcoma and collagen type I hydrogels in 2D (on the surface) and 3D (sandwich) conditions are the most common strategies likely affecting the accessibility of commercial ECM and easy-to-do methodologies [7,55]. Additionally, there is another alternative consisting of the use of ECM hydrogels: the extraction of ECM peptides by digesting decellularized liver matrices with pepsin in acidic solutions [57,58]. Interestingly, it was demonstrated that the composition of ECM extracted from decellularized livers may be different if the ECM is extracted at specific digestion times [59,60]. Early digestions (approximately 2 h) are enriched with peptides derived from less reticulated matrices such as the basement membrane, while late digestions (>4 to 72 h) contain high amounts of fibrillar collagen peptides.

Unfortunately, there may be a low reproducibility in ECM-based hydrogels because of the intrinsic variability of the purification processes [61]. Elastic and viscoelastic polymers are an excellent alternative to ECM hydrogels due to their reproducibility, tunability, and the possibility of conjugating any ECM component [62,63,64]. Two independent research groups cultured mouse hepatocytes on elastic polymers; both groups demonstrated that stiffness controls the transformation of primary hepatocytes in vitro: soft culture conditions preserved several functions in primary hepatocytes [25,26]. Many regulatory mechanisms caused by the impact of elasticity and viscoelasticity on primary hepatocytes are currently under investigation, and new culture platforms have been developed recently [32,37,65,66,67,68]. Viscoelasticity regulates attachment and spreading in primary hepatocytes, whereas the confinement and reduction of culture volumes have a strong impact on hepatocytes transformation [32,33,37,65]. Nevertheless, most of these attempts only opted for the of use collagen type I to promote cell adhesion, and the combination of elastic/viscoelastic polymers with an ECM that is more related to liver matrices has not been analyzed. There is an interesting report where ECM extracted from decellularized livers is incorporated into microfluidic chips, where both mechanical and biochemical cues are present at the same condition [67]. Therefore, it is necessary to evaluate the coexistence of mechanical and biochemical regulatory signals, allowing cell biologists and physiologists to better biomimetic microenvironments.

Here, we aimed at analyzing the role of collagen type I and basement membrane-associated matrix on rat primary hepatocytes cultured on soft conditions. We used commercial rat tail collagen type I and basement membrane extracted from EHS mouse sarcoma as controls of fibrillar collagens and epithelial basement membrane (BM) respectively, on polyacrylamide hydrogels (PAA HGs) with soft (1 kPa) and stiff (20 kPa) elastic moduli. To determine the effects of both matrices on cell culture, we combined 50% and 75% of collagen with BM or used decellularized liver extracellular matrices (dECM) from 2 h and 48 h of digestion. Our data demonstrated that hepatocytes adhere to stiff substrates independently of the ECM that is conjugated to the substrate. In contrast, BM had little effects on cell adhesion and aggregation in hepatocytes cultured on soft PAA HGs. Actin cytoskeleton remodeling was inhibited on soft conditions regardless of the conjugated ECM. It was demonstrated that the survival and aggregation of the primary hepatocytes are promoted by collagen type I on soft conditions. Regardless of the effects regulated by collagen and softness in primary hepatocytes when we explored metabolic functions, only the synthesis of glycogen was preserved in long cultures. This investigation highlights the important role that fibrillar collagens play in hepatocytes in soft conditions and suggests new culture conditions that allow us to explore the influence of ECM composition and mechanotransduction signals.

## 2. Materials and Methods

### 2.1. Materials

Rat tail collagen type I and basement membrane (Matrigel^®^) were obtained from Corning Life Sciences (New York, NY, USA). Recombinant epidermal growth factor (EGF) was obtained from SB Sino Biological (Beijing, China). Collagenase, heparin, Coomassie Brilliant blue R-250, and 2-Hydroxy-4′-(2-hydroxyethoxy)-2-methylpropiophenone (Irgacure 2959) were purchased from Sigma-Aldrich (St. Louis, MO, USA). Anti-phospho-p44/42 (that detects phosphorylated extracellular signal-regulated kinases 1/2 (ERK1/2)) and anti-p44/42 (that detects total ERK1/2) rabbit polyclonal antibodies were obtained from Cell Signaling (Danvers, MA, USA). Acrylic acid-N-hydroxysuccinimide ester, as well as anti-albumin and anti-ColA1 (B-10) antibodies were obtained from Santa Cruz Biotechnology (Dallas, TX, USA). Secondary anti-rabbit and anti-mouse HRP-conjugated antibodies and secondary Alexa Fluor 594-coupled anti-goat were purchased from Jackson ImmunoResearch (West Grove, PA, USA). Alexa 488-coupled phalloidin and DAPI (4′,6-diamidino-2-phenylindole) were purchased from Molecular Probes, ThermoFisher Scientific (Waltham, MA, USA). Culture media and reagents were obtained from Gibco, ThermoFisher Scientific.

### 2.2. Hepatocytes Isolation and Culture

The Institutional Animal Care and Use Committee of Instituto de Fisiología Celular and Facultad de Ciencias (UNAM) approved all of the animal protocols according to NOM 062-ZOO-1999 with the number PI_2019_02_004 approved in 27th February 2019. Hepatocytes were isolated from the livers of young Wistar rats (250 g) by using a collagenase perfusion method described in [69]. After isolation, hepatocytes were separated by centrifugation at 400 rpm for 2 min and washed four times with William’s E medium (Sigma-Aldrich). Viable hepatocytes were separated by Percoll (GE Healthcare, Chicago, IL, USA), and cell viability was evaluated using trypan blue exclusion. Fresh hepatocytes were suspended in attachment medium (DMEM-F12 supplemented with GlutaMAX, 1 mM sodium pyruvate, 10 mM HEPES, 0.5 μg/mL amphotericin B, insulin–transferrin–sodium selenite and 10% fetal bovine serum (FBS) plus antibiotics). Hepatocytes were seeded at a density of 2.5 × 10^5^ cells and allowed to adhere for 2–4 h on 20 mm round coverslips with 1 kPa and 20 kPa polyacrylamide hydrogels attached to the surface. Hepatocytes seeded on polystyrene plates covered with 1 mg/mL collagen type I dissolved in 20 mM acetic acid were used as cells cultivated on stiff standard conditions. Then, hepatocytes were maintained in a feeding medium (FBS-free attachment medium) in standard culture conditions of 5% CO_2_ at 37 °C. Differential interference contrast (DIC) microscopy images were captured by a Nikon Eclipse T*i*-S system.

### 2.3. Liver Decellularization and Pepsin Digestion

Decellularized extracellular matrices (dECM) were obtained from young Wistar rats as previously commented [57]. Briefly, anesthetized rats were subjected to a ventral laparotomy, and 0.2 mL of 1000 U/mL heparin were injected into the inferior vena cava; then, the portal vein was cannulated, and the liver was perfused with pre-chilled degassed deionized water at 5 mL/min for 30 min. Afterwards, washed livers were perfused with degassed decellularization solution (1% Triton-X 100, 0.1% NH_4_OH, (10 µg/mL) bovine pancreatic DNAse I and (1 mM) MgCl_2_), and it was recirculated overnight at room temperature (RT). Decellularized matrices were completely washed by perfusing them with deionized water for at least 4 h to ensure removal of the cellular components. dECM were stored at −70 °C before lyophilization. Lyophilized dECM were pulverized by fracturing them with liquid NO_2_. Then, 10 mg of pulverized dECM were digested with 1 mg of bovine gastric pepsin in a [0.1 N] HCl acidic solution in agitation at RT. dECM peptides were recovered at 2 and 48 h after digestion, and extracts were neutralized with (10 N) NaOH. Neutralized samples were centrifuged at 13.300 rpm for 10 min at 4 °C to discard insoluble fraction, and supernatants were recovered and stored at −70 °C before use.

### 2.4. Polyacrylamide (PAA) Hydrogels (HGs)

Polyacrylamide hydrogels were polymerized as described previously [70], and elastic properties were confirmed previously by microindentation [71]. Briefly, specific amounts of 40% acrylamide and 2% bis-acrylamide were mixed and deposited on 20 mm glass round coverslips, treated previously with (3-Aminopropyl)triethoxysilane (APTES)/glutaraldehyde as mentioned [72]. Then, 10% ammonium persulfate (APS) and 1% TEMED were added to generate persulfate radicals to polymerize PAA HGs for at least 30 min at RT. After polymerization, acrylic acid-NHS ester (1.7 mg/mL) and Irgacure 2959 (11 mg/mL) were added to rat tail collagen type I, Matrigel^®^, collagen/BM preparations and dECM peptides for conjugation to PAA HGs by exposing mixtures with a 365 nm wavelength with a nominal power density of 3.3 mW/cm2 during 3 min (UVP cross-linker CL-1000L). The total concentration used for conjugation was 0.1 mg/mL for all conditions as suggested in [73]. ECM-conjugated PAA HGs attached to coverslips were sterilized with 1× Dulbecco’s phosphate-buffered saline (DPBS) plus antibiotics before seeding.

### 2.5. SDS Polyacrylamide Gels Electrophoresis (SDS-PAGE) and Protein Staining

Protein quantification was performed by using the Protein Assay kit from Biorad according to the manufacturer’s instructions. Then, 20 µg of collagen type I, Matrigel, and collagen/BM combinations were loaded in 8–15% polyacrylamide (29:1) gels for SDS-PAGE. For silver staining, polyacrylamide gels were incubated in a fixation solution (50% methanol, 12% glacial acetic acid, and 0.5% formaldehyde) for 1.5 h followed by two washes of 2 min with deionized water. After fixation, PAA gels were washed three times in a 50% methanol solution for 10 min and rinsed with distilled water in between. Then, an oxidation reaction was performed by incubating with 0.004% Na_2_S_2_O_3_ for 35 s with two subsequent washes of deionized water for 20 s. An impregnation reaction was carried out using a solution containing 0.2% of AgNO_3_ plus 0.75% formaldehyde for 3 min afterwards with two subsequent washes of deionized water. Development was stopped at 10 s maximum in a solution of 6% Na_2_CO_3_, 0.004% Na_2_S_2_O_3_ plus 0.5% of formaldehyde; then, gels were washed twice for 10 s with deionized water. The reaction was completely stopped by adding the fixation solution for 30 min. All the reactions were performed in agitation at RT. For Coomassie staining, PAA gels were incubated in Coomassie blue solution (0.25% Coomassie R-250, 90% methanol/deionized water 1:1 and 10% acetic acid) overnight. After staining, PAA gels were washed and destained with the same solution (without Coomassie Brilliant blue) by replacing it several times until protein bands appeared.

### 2.6. Immunoblotting

For protein detection, hepatocytes cultured on PAA HGs were rinsed once with pre-chilled 1× DPBS and subsequently lysed with 0.5 mL of RIPA buffer (50 mM Tris-HCl pH 7.4, 150 mM NaCl, 1 mM ethylenediaminetetraacetic acid (EDTA), 0.5% sodium deoxycholate, 0.1% SDS, and 1% Nonidet) plus Complete^TM^ protease and phosphatase inhibitors cocktail in agitation for 1 h at 4 °C. Lysates were centrifuged at 13.300 rpm for 10 min, and supernatant was collected. Protein quantification was carried out afterwards, and 50–75 μg of protein were loaded in and separated by SDS-PAGE and transferred to a polyvinylidene fluoride (PVDF) membrane. Membranes were blocked for 1 h with 5% low-fat milk in TBS-Tween (Tris-HCl pH 7.4, 150 mM NaCl and 0.1% Tween-20). Primary antibodies against phospho-p44/42 (1:2000) and p44/42 were incubated overnight in TBS-Tween at 4 °C in agitation. For collagen detection, antibody against collagen type I (ColA1) was diluted 1:2000. Horseradish peroxidase (HRP)-coupled secondary antibodies against the light chain of immunoglobulin G (IgG) from mouse and rabbit were incubated in a dilution of 1:10000 for 1 h in blocking conditions at RT. Proteins were detected by using enhanced chemiluminescence assay (ECL, Merck Millipore, MA, USA).

### 2.7. Immunofluorescence and TUNEL Assay

Immunofluorescence and terminal deoxynucleotidyl transferase dUTP nick end labeling (TUNEL) assays were performed in rat primary hepatocytes cultured on PAA HGs as described in [74]. Briefly, hepatocytes were washed once with pre-warmed 1× DPBS and immediately fixed with 4% paraformaldehyde (PFA) in DPBS 1× at 37 °C for 20 min. Fixed cells were permeabilized with 0.1% Triton X-100 in 1× DPBS for 10 min in agitation at RT. After permeabilization, samples were blocked with 10% horse serum in 1× DPBS for 1 h in agitation at RT and then, primary antibody against albumin (1:1000) was incubated overnight in agitation at 4 °C in blocking conditions. Secondary antibody Alexa594-coupled anti-goat (1:500) was used for primary antibody detection. For the detection actin filaments, Alexa 488-coupled phalloidin was used at a dilution of 1:250 in 1× DPBS and incubated for 30 min at RT. Nuclei were stained with DAPI (1:200) in 1× DPBS and samples were mounted with Mowiol for preservation before imaging. Samples were imaged by epifluorescence microscopy (Eclipse C*i*-L, Nikon, Tokyo, Japan) and confocal microscopy (TCS-SP8, Leica, Wetzlar, Germany). Images were edited by using free Fiji software. For TUNEL detection, hepatocytes were fixed with 4% PFA and post-fixed in a pre-chilled 2:1 ethanol and acetic acid solution 5 min at −20 °C. After fixation, samples were processed according to the manufacturer’s instructions described in the ApopTag fluorescein in situ apoptosis detection kit manual (Merck Millipore).

### 2.8. Periodic Acid-Schiff (PAS) Staining

Fresh hepatocytes were seeded in 12-well plates and left to grow for one and five days; then, they were fixed at 37 °C with 4% paraformaldehyde (PFA) for 15 min. Following washing with 1× Dulbecco’s phosphate-buffered saline (DPBS) solution, hepatocytes were incubated with 1% periodic acid solution for 5 min, then stained with Schiff’s reagent (Sigma-Aldrich) for 15 min. All staining steps were carried out at room temperature except for fixation. Hepatocytes were rinsed with 1× DPBS after each step. Around 4–5 images of each growth condition were acquired with an Eclipse T*i*-S inverted microscope (Nikon Tokyo, Japan) using the 20× objective.

### 2.9. Quantification and Statistics

Image analysis and processing was done using ImageJ software (National Institutes of Health, Bethesda, MD, USA). We used a minimum of 5 images per sample, and all conditions were always performed in duplicate (10 images per condition, at least). For all data presented, there are at least 2 independent experiments. The statistical analyses were performed using GraphPad Prism 8 software. Data were statistically analyzed using one-way ANOVA and Tukey comparison tests. For statistical significance, a *p* < 0.05 was considered significant.

## 3. Results

### 3.1. Stiffness Promotes Cell Aggregation and Triggers Spreading in Primary Hepatocytes, Whereas Collagen Type I Promotes Cell Adhesion and Aggregation in Soft Conditions

It was recently reported that mouse primary hepatocytes respond to stiff PAA hydrogels (with an elastic modulus ≥10 kPa) by doubling their area in the first 12 h after seeding; however, this effect seems to be attenuated in viscoelastic PAA HGs [26,37]. To evaluate our conditions and confirm previous results, we decided to use soft (1 kPa, elastic modulus) and stiff (20 kPa, elastic modulus) polyacrylamide hydrogels conjugated with rat tail collagen type I as described in [74]. We evaluated the spreading of primary hepatocytes at 24 and 72 h after culture (Figure 1A). As expected, hepatocytes showed a significant difference in cell area when cultured on soft and stiff PAA HGs after 24 h of culture (Figure 1B). Hepatocytes did not modify significantly their cell area on soft substrates after 72 h of culture, whereas there was a large increase of cell area on 20 kPa PAA HGs (72 h, 1493 ± 363 µm^2^) when compared at the initial 24 h (2824 ± 870 µm^2^) of culture (Figure 1B). The composition of the perisinusoidal ECM is a combination of fibrillar collagens and proteins associated with BM such as collagen type IV [10,14]. It has been demonstrated that BM regulates differentiation and function in hepatocytes, although BM is located differentially along the hepatic lobule [21,28]. Therefore, we evaluated commercial rat tail collagen type I and BM (Matrigel^®^) at different combinations: 100% of collagen, 75% of collagen mixed with 25% of BM (75/25), and 50% of collagen mixed with 50% of BM (50/50) and 100% of BM (Figure 1C). The increasing amounts of collagen were confirmed by immunoblotting. Collagen type I was absent in BM; therefore, it was possible to analyze the effects of this fibrillar collagen on soft conditions. All conditions were conjugated at a concentration of 0.1 mg/mL in soft and stiff hydrogels, as reported for mammary epithelial cells [73]. Primary hepatocytes adhered to all culture conditions in 1 kPa PAA HGs at 24 h of culture (Figure 1D). Regardless of hepatocytes adhesion in all conditions, we observed cell condensation in apoptotic bodies [23] when hepatocytes were cultured on 1 kPa PAA HGs conjugated with BM even at 24 h of culture (Figure 1F). The presence of apoptotic bodies correlated with a decreasing number of attached hepatocytes after 72 h of culture on soft PAA HGs conjugated with BM. As collagen type I was included at different concentrations, apoptotic bodies were less evident, and cell numbers seemed to remain constant (Figure 1D).

A remarkable finding was the effect of collagen type I and stiffness on the aggregation behavior of primary hepatocytes in culture. As shown in Figure 1D–E, hepatocytes organized in aggregates, similar to hepatic cordons, when collagen type I was present regardless of the concentration at 24 h of culture on 1 kPa PAA HGs; nonetheless, larger aggregates were established in 100% of collagen. Individual hepatocytes were abundant in soft hydrogels conjugated with BM (Figure 1E). By contrast, primary hepatocytes are mostly organized in aggregates when cultured on 20 kPa PAA HGs. As expected, primary hepatocytes transformed into mesenchymal-like cells in all ECM combinations on stiff hydrogels after 72 h of culture (Figure 1D). In addition to collagen/BM combinations, we also conjugated ECM extracted from decellularized livers to PAA HGs [57,58,67]. We extracted ECM-related peptides from decellularized livers ECM (dECM) after 2 and 48 h of digestion with pepsin in HCl. As mentioned in [59], it is likely that after 2 h of digestion, dECM extracts contained peptides derived from non-collagenous proteins such as laminins. By contrast, it is possible that after 48 h of digestion, probably enriched dECM extracts with peptides derived from collagen type I and other fibrillar collagens. Differences in the extraction of ECM peptides is based on the reticulation status (cross-linking) of native decellularized ECM. The digestion of ECM is mainly based on the activity of pepsin; however, the accessibility of the enzyme also plays an important role [58,59]. Pepsin begins to hydrolyze poorly reticulated ECM in the first hours, whereas accessibility to highly reticulated matrices is limited to the enzyme. Thus, material obtained after 2 h of digestion is enriched with less cross-linked ECM in decellularized livers found along the sinusoids [21]. Therefore, we conjugated 2 h and 48 h dECM to 1 and 20 kPa PAA HGs (Figure 1D). As predicted, apoptotic bodies were observed on 1 kPa PAA HGs that were conjugated with 2 h dECM as well as on soft hydrogels linked with BM (Figure 1F). On soft PAA HGs conjugated with 48 h dECM, hepatocytes aggregated in spheroid-like structures; we observed similar aggregates on 50/50 ECM, as shown in Figure 1G. It is likely that 2 h dECM shares similarities with commercial BM in which collagen type I was not detected (Figure 1C). All these data demonstrated that stiffness promotes cell spreading independently of the conjugated ECM; however, it was suggested that collagen type I plays an important role in supporting cell viability and aggregation in hepatocytes cultured on soft conditions.

### 3.2. Actin Cytoskeleton Remodeling Is Inhibited as the Nuclear Area Is Confined in Parallel in Primary Hepatocytes Cultured on Soft PAA HGs

The actin cytoskeleton is severely remodeled in hepatocytes that are cultured on standard polystyrene plates with deposited collagen type I, passing from cortical F-actin to tensile stress fibers connected directly to integrins [33,35]. Many attempts to preserve functional hepatocytes have as a reference the maintenance of actin filaments beneath plasma membrane and proximal to canaliculi-like ducts in hepatocytes [29,30,65]. Recently, it was shown that actin remodeling is a fundamental process that controls differentiation and function in hepatocytes [33,39]. Hence, we evaluated whether hepatocytes remodeled their actin cytoskeleton on soft and stiff PAA HGs. As we expected, F-actin was close to plasma membrane and proximal to canaliculi-like structures in hepatocytes cultured on soft PAA HGs, regardless of the linked ECM. By contrast, actin cytoskeleton was reorganized into stress fibers in mesenchymal-like hepatocytes that had been cultured on stiff hydrogels after 72 h of culture (Figure 2A). The formation of stress fibers, associated with cell spreading and contractility [50], could explain the increase in cell area in hepatocytes cultured on stiff PAA HGs (Figure 1A). A 3D projection built from confocal microscopy images showed clearly the formation of intercellular partial ducts that resembled hepatic canaliculi in hepatocytes cultured on 1 kPa PAA HGs (Figure 2C). Polarized hepatocytes typically show these canaliculi-like structures in vitro [29]. On the contrary, there is a complete absence of cortical F-actin or canaliculi-like ducts in hepatocytes cultured on 20 kPa PAA HGs (Figure 2C).

It is known that intracellular tension caused by elastic and viscoelastic external properties is associated with a direct strain over the nucleus with subsequent modifications in nuclear dimensions [75,76]. Tensile actin filaments (stress fibers) are directly connected to nuclear lamins and modify nuclear transcriptional programs [77]. Therefore, we analyzed the area of the nuclei in primary hepatocytes cultured on soft and stiff hydrogels 72 h after culture. Hepatocytes cultured on 20 kPa PAA HGs presented a significant increase of their nuclear area (approximately 2-fold) when compared with soft conditions regardless of the conjugated ECM (Figure 2B). This occurred in parallel with the increase of stress fibers in mesenchymal-like hepatocytes (Figure 2A). Remarkably, nuclei were confined in hepatocytes that adhered to soft PAA HGs after 72 h of culture (Figure 2B). In addition, there were significant differences in the nuclear areas of hepatocytes cultured on different ECM linked to 1 kPa PAA HGs; it seems that nuclear dimensions increased as collagen percentage augmented, as shown in Figure 2B. Again, hepatocytes cultured on soft PAA HGs linked to BM or 2 h dECM showed similar results; both conditions presented apoptotic bodies (Figure 1G). All data shown demonstrated that low elastic modulus (1 kPa) did not cause enough tensile forces that might promote actin cytoskeleton remodeling. The confinement of nuclei likely inhibited rearrangements in chromosome territories so that transcriptional programs could remain unaltered; thus, it is likely that hepatocytes cultured on soft PAA HGs might preserve important hepatic functions.

### 3.3. Collagen Type I Increases Viability in Primary Hepatocytes Cultured on Soft PAA HGs and Participates with Epidermal Growth Factor (EGF) in Regulating Cell Aggregation

As shown in Figure 1G, apoptotic bodies were identified when primary hepatocytes were cultured on soft PAA hydrogels linked to BM or 2 h dECM (Figure 1B). Morphological characterization is not sufficient to draw any conclusions about the effects that collagen type I may have on the viability of primary hepatocytes. Hence, we analyzed the number of hepatocytes that suffered DNA fragmentation on soft hydrogels by using TUNEL assay that detects double-strand DNA breaks [78]. Primary hepatocytes positive to TUNEL were significantly more abundant in 1 kPa PAA HGs when BM and 2 h dECM were conjugated, compared with collagen after 72 h of culture (Figure 3A,B). Apoptosis might have been triggered by the absence of proliferation and survival signals in this case [79]. It was indeed reported that stiffness regulates the activation (phosphorylation) of ERK1/2 proteins in hepatocytes [23,56]. Thus, we wondered whether collagen type I activated mitogen-activated protein kinases (MAPK), ERK1, and ERK2. The phosphorylation of ERK1/2 kinases did increase as collagen type I concentration augmented in hepatocytes cultured on soft PAA HGs after 24 h of culture (Figure 3C). Additionally, the activation of ERK1/2 proteins increased in hepatocytes cultured on stiff hydrogels linked to the 50/50 ECM. It is likely that collagen type I might start playing an important role as mechanical conditions softened. In addition, we analyzed whether epidermal growth factor (EGF) was able to rescue the viability of hepatocytes cultured on 1 kPa PAA HGs. EGF was able to reduce significantly the number of hepatocytes positive to TUNEL when 2 h dECM was conjugated. Nevertheless, EGF stimulation did not reduce the number of apoptotic cells in PAA HGs linked to BM (Figure 3B). Interestingly, there was a basal number of hepatocytes positive to TUNEL when cultured on soft hydrogels regardless of the nature of the conjugated ECM or the presence of EGF stimulation. However, whether this number is reduced in stiff hydrogels needs further investigation.

After analyzing survival, we evaluated the effects of collagen type I and EGF stimulation on the proliferation state of primary hepatocytes cultured on soft PAA HGs. Unexpectedly, none of the conditions tested presented a significant difference in the number of hepatocytes after 72 h of culture (Figure 3E). There was no significant increase in cell population by EGF stimulation on soft conditions regardless of the conjugated ECM. Nevertheless, we noticed that the aggregates of primary hepatocytes became larger when hepatocytes were stimulated with EGF (Figure 3D). By measuring the area covered by cell clusters, aggregates of primary hepatocytes were significantly larger when both collagen and EGF stimulation were present on soft PAA HGs after 72 h of culture (Figure 3F–G). However, this effect on cell aggregation was not observed when 48 h dECM was linked to soft hydrogels. Again, aggregates adhered on 48 h dECM showed a spheroid-like conformation (see Figure 1D). The data shown here demonstrated that collagen type I plays a role in the maintenance of the survival of hepatocytes cultured on soft conditions. As a remarkable finding, EGF stimulation seems to regulate aggregation in primary hepatocytes; this effect has not been reported in such stiffness control conditions.

### 3.4. Primary Hepatocytes on Soft PAA HGs Maintain their Actin Cytoskeleton Proximal to the Plasma Membrane, Canaliculi-Like Structures, and Synthesis of Glycogen, Whereas They Lose Albumin Protein Expression during Long Cultures

Both the elastic properties and the composition of ECM control differentiation and several cellular functions in primary hepatocytes during culture [25,26,28,55]. Desai et al. demonstrated that stiffness promotes integrin β1-dependent focal adhesions in stiff PAA HGs (60 kPa), resulting in the loss of hepatocyte nuclear factor 4 alpha (HNF4α) mRNA expression and consequently the loss of functional primary hepatocytes in culture. In that report, hepatocytes were cultured on very soft PAA HGs (approximately 140 Pa) during 24 h. Unfortunately, they did not show the organization of hepatic aggregates for more than 24 h of culture, being impossible to evaluate whether primary hepatocytes remained as individual cells or formed aggregates in such soft conditions, as shown in Figure 1E. In the liver, hepatocytes are naturally organized in cordons (elongated aggregates) so that individual hepatocytes could mislead us in the information generated from such conditions, as well as several important signals that exist only when cells aggregate and form unions might be lost.

Exploiting our culture conditions, we further studied the organization and maintenance of functional primary hepatocytes on soft PAA HGs with different linked ECM at 120 h of culture (5 days). As shown in Figure 4A, primary hepatocytes still maintained themselves in aggregated structures regardless of the nature of EGF stimulation on soft hydrogels. Nonetheless, EGF stimulation was necessary to the survival of hepatocytes cultured on 48 h dECM-conjugated 1 kPa PAA HGs; apoptotic bodies were indeed numerous in this condition after 120 h of culture without EGF stimulation. Remarkably, the nuclear area did not significantly increase in soft hydrogels at 120 h of culture when compared with nuclear areas observed in hepatocytes cultured for 24 h when collagen type I and 48 h dECM were conjugated (Figure 4C). Interestingly, EGF stimulation unlocked the confinement of nuclear dimensions when collagen type I was linked, regardless of its concentration (Figure 4D). In parallel, cortical actin filaments and actin proximal to canaliculi-like structures were preserved in all conditions tested after 120 h of culture (Figure 4B). It was the first time that actin cytoskeleton remodeling was inhibited in primary hepatocytes by culturing them on polymeric hydrogels with controlled mechanical properties.

Thus, we hypothesized that it was possible that some cellular functions were preserved in primary hepatocytes cultured on soft PAA HGs for 24 h of culture, as reported in [26]. We evaluated the expression of albumin protein and the synthesis of glycogen in hepatocytes at 24 and 120 h of culture (Figure 4B). In particular, albumin protein expression was detected at 24 h of culture but reduced after 120 h of culture. It was also noticeable that albumin protein was localized proximal to canaliculi-like structures (Figure 4B). These ducts have been reported as tensile structures that regulate proliferation and polarity [9,29]. Zeigerer et al. demonstrated that polarized primary hepatocytes confined early endosomes at cortical F-actin and actin filaments close to canaliculi-like structures, as we demonstrated for albumin protein. For glycogen synthesis, we performed the periodic acid-Schiff (PAS) staining to detect glycogen storages in the primary hepatocytes [80]. Surprisingly, glycogen staining presented a strong purple staining in hepatocytes cultured on 1 kPa PAA HGs (both 75% and 100% collagen) at 24 and 120 h after culture, whereas PAS reactivity was faint (soft purple) in hepatocytes cultured on stiff standard conditions, even after only 24 h of culture (Figure 4E). Glycogen storage is an important function of the hepatocytes, and its maintenance in soft conditions up to 120 h is a promising progress in preserving metabolic functions in primary hepatocytes. Overall, these results confirmed that soft elastic conditions halt the remodeling of actin cytoskeleton in primary hepatocytes, but it does not prevent the loss of some parenchymal functions. This needs further investigation to determine which functions are preserved in soft conditions and if some hepatic functions require additional signaling.

## 4. Discussion

Perisinusoidal space is constituted of fibrillar collagens and a basement membrane-related matrix organized in a reticular ECM that surrounds hepatic cordons and HSCs but is beneath LSECs [2,6]. ECM proteins are distributed differentially along the hepatic lobules, and there is a more intricate matrix that comes from portal and central tracts. Thus, hepatocytes sense different mechanical and biochemical properties, depending on their location [10,16,17,21,81]. It is likely that ECM distribution is associated with the zonation of hepatocytes along hepatic lobules [1,21]. To the best of our knowledge, in spite of the abundance of fibrillar collagens in direct contact with parenchyma, there has not been a serious effort to study the role of fibrillar collagens on primary hepatocytes within a soft mechanical context until now. Hence, we aimed our research to evaluate the impact of collagen type I, that together with collagen type III is one of the most abundant fibrillar collagens in the liver [14,15], in combination with a basement membrane-related matrix on rat primary hepatocytes cultured on soft polyacrylamide hydrogels. Herein, we confirmed that the transformation of primary hepatocytes into a mesenchymal-like phenotype is driven by stiffness, as we proved in 20 kPa PAA HGs. It seems that the generation of intracellular tensile forces, e.g., on stiff materials with a high elastic modulus, is crucial in the maintenance of functional hepatocytes [26,33]. However, force transmission might be modulated by anchoring options presented in culture [35,38] as it was demonstrated for mammary epithelial cells where ECM composition and 3D culture conditions affected cell response to the mechanical properties [73].

Here, we demonstrated that apoptosis was triggered in the absence of collagen type I. It is likely that fibrillar collagens participate in the enhancement of hepatocytes viability and the activation of other signaling pathways such as AKT kinases [23]. It is interesting to consider whether there is an additional mechanism that activates ERK1/2 proteins in hepatocytes cultured on stiff conditions due to the fact that we observed that ERK1/2 phosphorylation was increased in stiff hydrogels linked to 50/50 ECM [24]. It was unexpected that EGF stimulation did not rescue hepatocytes viability in soft hydrogels conjugated with BM, although it did downregulate apoptosis in 2 h dECM. Not only collagen concentration and EGF stimulation rescued hepatocytes from apoptosis, but also both of them demonstrated a strong effect on the formation of aggregates of primary hepatocytes in soft conditions. The more collagen linked to soft PAA HGs, the more aggregates were observed. On the other hand, EGF stimulation enhanced the aggregation of primary hepatocytes in 1 kPa PAA HGs conjugated with collagen/BM mixtures but not in 48 h dECM. It is known that integrin β1 is necessary for a proper growth during liver development [82], and blocking integrin β1 decreases myosin activity in hepatocytes and aggregation in hepatocytes [35,83]. This could explain the differences in cell aggregation observed in this work. In addition, EGF receptors activity could be influenced by focal adhesions, and this likely affects proliferation in hepatocytes [84,85]; nonetheless, longer culture times and DNA replication assays must be evaluated to discard the absence of proliferation. It will require further investigation to understand the interesting cross-talk between stiffness, ECM composition, and EGF signaling.

Remarkably, actin cytoskeleton was not significantly remodeled in primary hepatocytes during 120 h of culture in soft PAA HGs. The maintenance of polarity had not been reported for such polymeric substrates, as it has been demonstrated for ECM-based sandwich 3D cultures. Actin filaments were localized proximal to plasma membrane and to canaliculi-like structures. Recently, it was demonstrated that canaliculi are contractile structures that activate mechanotransduction pathways in liver regeneration [9]. The formation of canaliculi-like structures could be associated with the inhibitory effect that soft PAA HGs have on nuclear strain in primary hepatocytes shown by the increase of nuclear area in parallel with stress fibers appearance. Actin stress fibers, but not cortical actin fibers, connect directly to the nucleus through the linker of nucleoskeleton and cytoskeleton (LINC) complex [46,76]. It is likely that the transformation of hepatocytes to mesenchymal-like cells requires tensile forces connected to the nuclear envelope in order to strain and rearrange chromosome territories so that the maintenance of cortical F-actin and nuclear magnitudes might preserve transcriptional programs that sustain functions in the hepatocytes [33,39,77].

The effects of elastic and viscoelastic mechanical properties on primary hepatocytes has been recently reported [25,26,37]. These investigations have studied the mechanotransduction of primary hepatocytes from minutes to days and elastic modulus tested ranged from approximately 160 Pa to 3 kPa. In all cases, researchers demonstrated the maintenance of hepatic functions despite the differences in culture conditions. Sun et al. demonstrated that regardless of the external mechanical properties, confinement of the hepatocytes is enough to preserve several functions of hepatocytes during 3 weeks [33]. Therefore, it is possible that independently of the materials used in those reports, any culture condition that establishes cell confinement in primary hepatocytes will preserve a functional phenotype by inhibiting tensile forces mediated by stress fibers [33,51]. The regulation of mechanotransduction signaling seems to be relevant in the promotion or the attenuation of some physiopathological processes where ECM remodeling plays an important role by modifying mechanical and biochemical characteristics in the surrounding environment. In the particular case of fibrosis, it is likely that the activation of mechanotransduction pathways such as Yes-associated protein and transcriptional coactivator with PDZ-binding motif (YAP/TAZ) proteins is crucial to the development and progress of fibrosis. Interestingly, blocking mechanotransduction signaling attenuates the fibrogenesis events in a kidney fibrotic model [86]. In that study, mechanotransduction pathways were evaluated in parallel to fibrotic models, but the authors used stretching as a mechanical stimulation. It is likely that culture conditions performed in this work would be more suitable for testing mechanical impact for in vitro models.

A partial maintenance of polarity might let us study cell processes that highly depend on cell polarization such as vesicle transport and the formation of cellular junctions. Interestingly, albumin was predominantly localized in canaliculi-like structures. It was reported that the early endosome antigen 1 (EEA1) protein was also localized close to canaliculi-like structures in primary hepatocytes cultured on collagen sandwich platforms [29]. The downregulation of albumin protein expression after 120 h of culture even though hepatocytes do not remodel their actin cytoskeleton might be explained because albumin expression is regulated by other signals in addition to mechanics [87,88]. By contrast, the synthesis of glycogen remained in primary hepatocytes up to 120 h after culture. It seemed to be independent of conjugated ECM, although other ECM proteins will need to be tested probably in the presence or absence of EGF to avoid apoptosis. It has been reported that glycogen synthesis may be regulated by actin polymerization [89,90]. Thus, controlling actin cytoskeleton dynamics by culture conditions might be related to the preservation of glycogen synthesis. In summary, we aimed to identify optimal conditions for primary hepatocytes, considering both mechanical and biochemical characteristics. This investigation indicates the effects of ECM on soft cultures and opens the possibility to evaluate cell processes and signaling pathways in primary hepatocytes on biomimetic environments in order to understand as profoundly as possible the particular factors that strongly influence the phenotype of primary hepatocytes in vitro.

## 5. Conclusions

The presented work gathers previous observations of primary hepatocytes cultured in biomimetic cultures [25,26,27,28], focusing in controlling two important characteristics: ECM protein composition and stiffness. Data clearly demonstrated the contribution of both external features in the survival, organization and differentiation of the hepatocytes and their possible relevance in certain physiopathological processes in the liver. Particularly, it was shown the prominent role of collagen type I in the survival of the hepatocytes in controlled soft conditions (1kPa PAA HGs). In addition, collagen type I promotes aggregation in hepatocytes independently of EGF signaling. Nevertheless, EGF stimulation promotes larger aggregates. At the same time, ECM derived from decellularized livers (dECM) share similarities with basement membrane (BM)/collagen combined matrices but a further characterization is needed to determine a good dECM extraction for culturing primary hepatocytes. Finally, it was likely that hepatocytes preserved a set of metabolic and non-metabolic functions when cultured for long periods in soft conditions where develop stable aggregates. Nonetheless, it will require additional investigation to establish whether dECM or growth factor signaling might participate in the regulation of many other functions sustained by the hepatocytes in the liver.

## Figures and Tables

**Figure 1 biomimetics-05-00030-f001:**
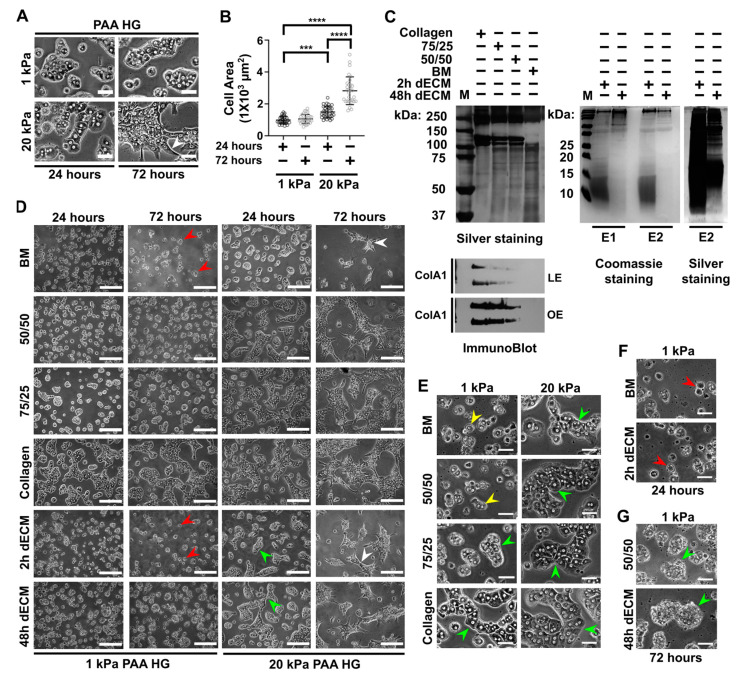
Collagen type I is necessary for cell adhesion and aggregation in rat primary hepatocytes cultured on soft polyacrylamide hydrogels (PAA HGs). (**A**) Magnifications of representative differential interference contrast (DIC) images of primary hepatocytes cultured on soft (1 kPa) and stiff (20 kPa) polyacrylamide hydrogels (PAA HGs) conjugated with collagen type I after 24 and 72 h of culture. Scale bar: 50 μm. (**B**) Quantitative analysis and comparison of cell area of hepatocytes cultured on soft and stiff PAA HGs at 24 h and 72 h of culture. Data are represented as mean ± standard deviation (SD) of a representative of 3 independent experiments, **** *p* < 0.0001, *** *p* < 0.001 (**C**) Silver staining of SDS-PAGE of Matrigel basement membrane-associated matrix (BM) and collagen type I at different proportions (left) and decellularized liver matrices (dECM) extracted after 2 h and 48 h of digestion (right). Lower panel: collagen type I immunoblotting in collagen and BM mixtures (below). Membranes low-exposed (LE) and over-exposed (OE) (lower panel). In addition, Coomassie staining of 2 h and 48 h dECM of 2 independent experiments (E1 and E2) ran in parallel. (**D**) Primary hepatocytes cultured on 1 kPa and 20 kPa PAA HGs conjugated with different collagen I/BM proportions and decellularized matrix extracts (dECM) at 24 h and 72 h of culture. Images were obtained from a 20× objective. Scale bar: 200 μm. (**E**) Magnifications of differential interference contrast (DIC) images of primary hepatocytes cultured on soft and stiff PAA HGs at 72 h of culture. Collagen and BM were conjugated at different proportions. Scale bar: 50 μm. (**F**–**G**) Magnifications of DIC images of hepatocytes cultured on soft PAA HGs conjugated with different extracellular matrices (ECMs) at 72 h of culture. ECM mixtures: BM, 100% matrigel; 50/50, 50% collagen type I plus 50% BM; 75/25, 75% collagen type I plus 25% BM; collagen, 100% collagen type I; 2h dECM, dECM after 2 h of digestion; 48h dECM, ECM after 48 h of digestion; M, molecular weights. Scale bar: 50 μm. Aggregates of primary hepatocytes: green arrowheads; apoptotic bodies: red arrowheads; individual hepatocytes: yellow arrowheads; transformed hepatocytes: white arrowheads.

**Figure 2 biomimetics-05-00030-f002:**
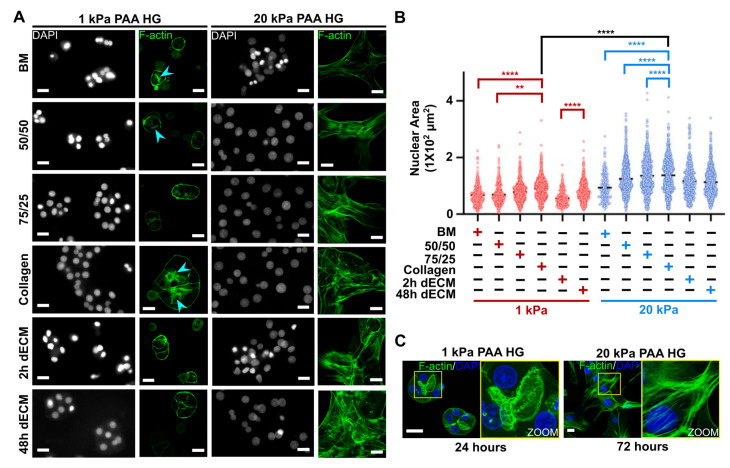
Actin cytoskeleton remodeling and nuclear area increase are inhibited in primary hepatocytes cultured on soft PAA HGs. (**A**) Representative fluorescence images of nuclei (4′,6-diamidino-2-phenylindole, or DAPI) and actin filaments (F-actin) in hepatocytes after 72 h of culture on soft and stiff PAA HGs conjugated with ECM matrices. Actin filaments proximal to canaliculi-like structures: light blue arrowhead. Images were obtained from 40× (epifluorescence) and 63× (confocal microscopy) objectives for nuclei and F-actin, respectively. Scale: 20 µm. (**B**) Quantitative analysis of nuclear area in primary hepatocytes cultured on soft and stiff PAA HGs linked to different ECM. ECM mixtures: BM, 100% Matrigel; 50/50, 50% collagen type I plus 50% BM; 75/25, 75% collagen type I plus 25% BM; collagen, 100% collagen type I; 2 h dECM, dECM after 2 h of digestion; 48 h dECM: ECM after 48 h of digestion. Significant differences are shown in red and blue between PAA HG with the same elastic condition; black lines show significant differences between PAA HGs with the different elastic conditions. Data are represented as mean ± SD of a representative of two independent experiments, ** *p* < 0.01, **** *p* < 0.0001. (**C**) 3D projections of confocal images of hepatocytes cultured on 1 kPa and 20 kPa PAA HGs at 24 h and 72 h after culture, respectively. Canaliculi-like structures and stress fibers are highlighted and magnified at yellow squares (right panels). Scale bars: 20 µm.

**Figure 3 biomimetics-05-00030-f003:**
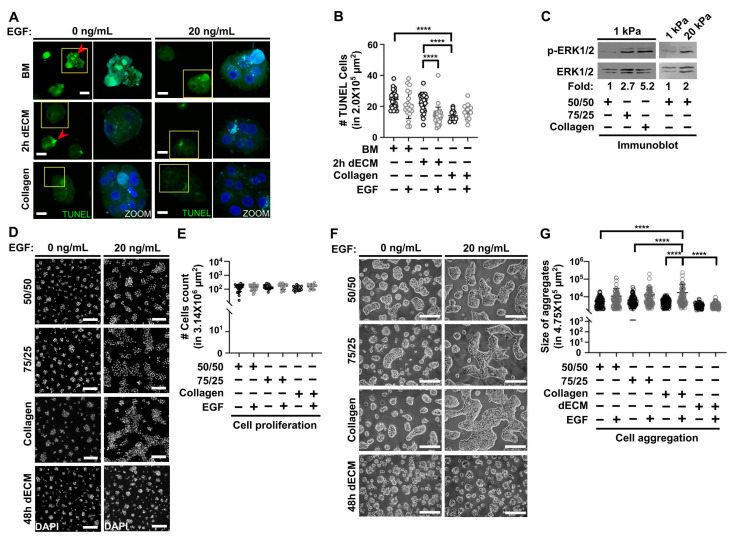
Collagen type I and epidermal growth factor (EGF) stimulation regulates cell viability and aggregation in primary hepatocytes cultured on soft PAA HGs. (**A**) 3D projections of representative fluorescence confocal microscopy images of hepatocytes positive to TUNEL in soft PAA HGs conjugated with collagen, BM and 2 h dECM in the absence or presence of recombinant EGF (20 ng/mL) after 72 h of culture. Apoptotic hepatocytes with double-strand DNA breaks: red arrowheads. Images were obtained from a 40× objective. Scale bar: 20 μm. (**B**) Quantification of positive TUNEL cells in soft PAA HGs conjugated with collagen I, BM, or 2 h dMEC in the absence or presence of EGF [20 ng/mL]. Data are represented as the mean ± SD of 2 independent experiments; **** *p* < 0.0001. (**C**) phospho- extracellular signal-regulated kinases 1/2 (ERK1/2) and ERK1/2 immunobotting in hepatocytes cultured on 1 kPa and 20 kPa PAA HGs conjugated with different ECM after 24 h of culture. Data presented are representative of 2 independent experiments. Numbers below blots indicate the densitometric ratio values of phospho-ERK1/2 and total ERK1/2 protein, expressed in fold changes. (**D**) Representative fluorescence images of nuclei (DAPI) in hepatocytes after 72 h of culture on soft PAA HGs conjugated with different ECMs in the absence or presence of EGF (20 ng/mL). Images obtained using a 10X objective. Scale bar: 200 μm. (**E**) Cells count graph of conditions depicted in (**D**). (**F**) Representative DIC images of aggregates of primary hepatocytes cultured on 1 kPa PAA-HGs linked to different ECMs in the absence or presence of EGF (20 ng/mL) at 72 h of culture. Images were obtained from a 20x objective. Scale bar: 200 μm. (**G**) Quantification of size of cell aggregates in the conditions depicted in (**F**). ECM mixtures: BM, 100% Matrigel; 50/50, 50% collagen type I puls 50% BM; 75/25, 75% collagen type I plus 25% BM; collagen, 100% collagen type I; 2 h dECM, dECM after 2 h of digestion; 48 h dECM: ECM after 48 h of digestion. Data are represented as mean ± SD of 2 independent experiments; **** *p* < 0.0001.

**Figure 4 biomimetics-05-00030-f004:**
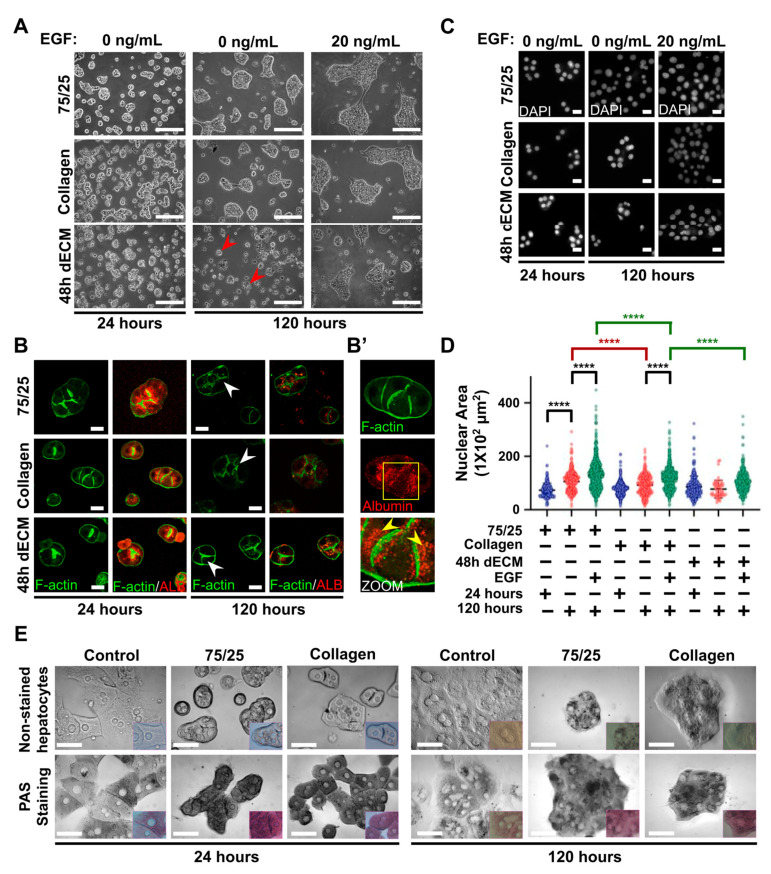
Cortical actin filaments and actin proximal to canaliculi-like structures are preserved in primary hepatocytes cultured on soft PAA HGs in long cultures, but albumin protein expression is lost. (**A**) Representative DIC images of hepatocytes cultured on soft PAA HGs linked different ECM at 24 h and 120 h of culture in the absence or presence of (20 ng/mL) EGF after 24 and 120 h of culture. Apoptotic bodies: red arrowheads. Images were obtained from a 20× objective. Scale bar: 200 μm. (**B**) Representative immunofluorescence confocal microscopy images of albumin protein and F-actin in hepatocytes cultured on 1 kPa PAA HGs conjugated with different ECMs at 24 h and 120 h of culture. Canaliculi-like ducts: white arrowheads. Images were obtained from a 63× objective. Scale bar: 20 μm. (**B’**) 3D projection from confocal microscopy images of F-actin and albumin protein in primary hepatocytes cultured on soft PAA HGs at 24 h of culture. The localization of albumin protein is highlighted in yellow squares. Albumin protein distribution: yellow arrowheads. (**C**) Representative epifluorescence images of nuclei (DAPI) of hepatocytes cultured on soft substrates conjugated different ECM after 24 h and 120 h of culture in the absence or presence of (20 ng/mL) EGF. Images were obtained from a 63× objective. Scale bar: 20 μm. (**D**) Quantification of nuclear areas of hepatocytes cultured on conditions depicted in (C). Data are represented as mean ± SD of a representative of 2 independent experiments, **** *p* < 0.0001. (**E**) Periodic acid-Schiff (PAS) staining detection showed that glycogen storage is abundantly conserved in primary hepatocytes cultured on soft PAA hydrogels compared with those cultured on standard polystyrene plates with deposited collagen. Upper panels: show non-staining hepatocytes cultured for 24 through 120 h. Lower panels: Cytoplasm with darker gray shades indicates higher glycogen storage levels. Inner panels show color images of both stained and non-stained hepatocytes. Purple-magenta color indicates positive cells to glycogen. Representative images were collected from each of three independent experiments per treatment condition. ECM mixtures: collagen, 100% collagen type I; 75/25, 75% collagen type I plus 25% BM; 48 h dECM: ECM after 48 h of digestion. Scale bars; 50 μm.

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
