# Peer review of "Fibrillar Collagen Type I Participates in the Survival and Aggregation of Primary Hepatocytes Cultured on Soft Hydrogels"

_biomimetics, 2020, doi:10.3390/biomimetics5020030_

Round 1
Reviewer 1 Report
The overall scope of this manuscript is interesting and of value to the field. The aim of the study was to identify optimal conditions for culturing primary rat hepatocytes in terms of mechanical and chemical characteristics of the surrounding ECM. The authors nicely showed that cell morphology and viability is influenced by the stiffness of the substrate and the ECM components that have been attached to it.
The manuscript covers many different conditions, which makes it difficult to keep track and at some point confused me. Although I don't see how this can be improved, I think this is worth mentioning. May be the authors can make some structural changes to make this more clear.
Although many conditions have been tested, the authors show limited functionality of the hepatocytes. Albumin expression alone, assessed by immunofluorescent stainings, not a real quantitative measurement. It would be interesting if they would also show how the functionality of hepatocyte is affected by the different physical and biochemical culture environment. This could for example be shown on mRNA level by genetic analysis with qPCR, by doing an albumin ELISA, or assessing specific CYP enzymes. Such data would greatly attribute to the statement the authors try to make.
In the introduction, the authors discuss the use of ‘soft hydrogels’ or ‘soft conditions’. However, they do not give a clear insight what they regard as being a soft material. The use of the term ‘soft’ or stiff’ in this particular setting is not clear to me. It would be helpful is these terms are better defined in the introduction. Furthermore, it would be good to compare the physical characteristics of the hydrogels to those of native liver tissue. In addition to this, I would suggest to add the mechanical characterization of the different conjugated hydrogels used. Does the addition of ECM components alter the mechanical strength of the hydrogels?
The remark that it is likely that the 2h ECM digest is more comparable to commercial BM is unclear to me. Does the 2hour ECM digest result in similar levels of Laminin-111 when compared to commercially available BM? Or is there any other proof to show that the two are similar? In addition to that, how does the 2hour ECM digest differ from the 48hour ECM digest? I was unable to deduce this from the reference given by the authors. Obviously, there will also be a difference in digestion-efficiency. Could the authors explain a bit more on what the differences in digestion efficiency were? And how they correct for that, in order to ensure that similar protein concentrations were used to conjugate the hydrogels with?
A minor comment which can relatively easily be adjusted is the clarity of the figures. I think the resolution of the figures is way too low to see details well enough. The panels need to be larger in order to appreciate the content.
Another minor comment is about the following sentence in the aims section of the introduction: ‘Intriguingly, albumin expression was lost despite actin cytoskeleton organization was maintained during long cultures.’ This sentence does not make much sense as it is right now. Please rephrase.
Reviewer 2 Report
Serna-Márquez and coworkers report on a study that compares the rearrangements in the ECM composition and mechanical properties with the final aim to assess the role of fibrillar collagens and basement membrane-associated matrix on actin cytoskeleton and function of rat primary hepatocytes cultured on soft elastic polyacrylamide hydrogels.
The methods are sound and straight forward, and have been well described however some additional information may be required. The study appears well conducted and the paper is written comprehensibly, although the discussion part could be a bit more stringent.
Main concerns:
On the PAAHGs fabrication the authors mentioned the use of Irgacure 2959 and UV light to crosslink the hydrogels. However, the authors should state the energy dose used for crosslinking and in the same time to specify the difference between the soft and stiff hydrogels (1kPa and 20kPa) i.e. chemical composition or exposure time.
In this study, the authors focused on the cell viability and aggregation, actin structure and remodelling and variation of ECM used to culture the rat hepatocytes. However, the several important aspects have not been evaluatedor discussed in the manuscript such as tight junction formation, functional metabolic readouts of the hepatic cells i.e. albumin, CYP, AST, ALT, urea, glucose etc, and the effect of soft and stiff hydrogels on epithelial–mesenchymal transition, which has been confirmed to have an effect on the hepatic epithelial cells.
Another aspect that authors should consider is the effect of the collagen in the PAAHGs with regards of the integrin binding sites for cell attachment and proliferation. This could be a better way, among others, to explain the results regarding the formation of more aggregates on soft PAAHGs with higher collagen in composition. This should be discussed at least as a potential explanation for the aggregation behaviour when collagen is added to the PAAHGs as well as other potential hypothesis.
For the reference section, the authors should modify the reference 70 in accordance with the journal style.
Reviewer 3 Report
Serna-Marquez et al. sought to identify the morphological and phenotypic responses of primary rat hepatocytes to fibrillary- or basement membrane-associated matrix (BM) proteins. This was done through the absorption of proteins in various combinations to soft elastic polyacrylamide gels of 1- or 20-kPa Young’s moduli in order to replicate the stiffening effects of the liver during organogenesis or regeneration. Authors evaluated morphologies on underlying compositions of interest and found 1) fibrillary proteins support ‘hepatic cordon’ formation’, especially on soft substrates, whereas stiff substrates increase aggregation and cell area (most spreading found after 72 hours), 2) canaliculi-like ducts formed via cortical F-actin found on soft substrates, only, 3) collagen supports viability on a soft substrate as shown in reduced TUNEL staining, 4) epidermal growth factor (EGF) supports aggregation of hepatocytes and improved viability on 48 h dECM, and 5) ECM proteins conjugated to soft substrate supports polarity but does not rescue albumin (immunostaining) after five days of culture. The following issues should be addressed to improve the manuscript prior to publication.
Major Issues:
1. Authors state that “it is likely that after 2 h of digestion dECM extracts contained peptides from BM [and] after 48 h of digestion dECM extracts are probably enriched with peptides from collagen type I and other fibrillary collagens,”; however, both of these conditions are not evaluated for silver staining nor immunoblot. Thus, authors should re-evaluate these conditions to support the speculation over the composition of dECM at various digestion lengths.
2. Authors need to quantitatively measure commonly used hepatocyte markers, such as albumin production via ELISA, urea synthesis, and cytochrome P450 enzyme activity in order to demonstrate the differential loss due to either ECM and/or underlying matrix stiffness to support claims of Figure 4.
3. The authors must comment on the potential implications of a disease-like context being replicated in vitro on their findings, given that the increase in fibrillary collagens and/or stiffness is reminiscent of the progression of fibrosis.
Minor Issues:
1. In general, it is unclear why only certain conditions are down selected for analysis, especially in Figure 3A, D, and therefore, authors should clarify their rationale in the text.
2. Fig. 1D: The authors assess collagen content using immunoblot in their collagen and Matrigel mixtures. It would be beneficial to assess other ECM proteins that would be found in the Matrigel to fully understand the compositions of their ECM mixtures.
3. Fig. 1: Colored arrowheads make it difficult to appraise in black-and-white print, thus, it is suggested that authors utilize different arrowheads to display important features of the figure in addition to different colors.
4. Figure 2: The authors quantify the nuclear area in the various ECM and stiffness conditions. Additionally, the authors mentioned that the BM and 2h dECM conditions had many apoptotic bodies. Did the authors specifically remove these apoptotic bodies from their nuclear area analysis, since including the apoptotic bodies could skew the results. Please provide a more thorough description of image analysis in methods.
5. Figure 2C: The authors used an actin filament stain to conclude that they have canaliculi-like structures. The authors should use a stain specific for bile canaliculi to support this claim.
6. Fig. 3: Panel F is missing description in the legend and it is unclear form the figure of the dECM represented is from 2 or 48 h, though the text suggests the latter.
7. Fig. 3C: The authors claim that the phosphorylation ratio of ERK 1/2 is higher with increased collagen content; however, overall ERK 1/2 seems to be higher in collagen conditions rather than an increased ratio of p-ERK 1/2 to ERK 1/2. Densitometry analysis with appropriate statistics will be more appropriate for conclusions of pathway activity.
8. The manuscript requires minor editing with respect to spelling, grammar, syntax, word usage, etc.
Round 2
Reviewer 1 Report
Considering the response to my initial review and adaptations made to the manuscript, I don't see major issues.
Reviewer 3 Report
the authors have adequately addressed all of my comments. While I would have liked to see the CYP3A4 data in the main manuscript, I agree that its more thorough investigation is reserved for a future manuscript